# Clinical Ketosis-Associated Alteration of Gene Expression in Holstein Cows

**DOI:** 10.3390/genes11020219

**Published:** 2020-02-19

**Authors:** Zhou-Lin Wu, Shi-Yi Chen, Chao Qin, Xianbo Jia, Feilong Deng, Jie Wang, Song-Jia Lai

**Affiliations:** 1Farm Animal Genetic Resources Exploration and Innovation Key Laboratory of Sichuan Province, Sichuan Agricultural University, Chengdu 611130, China; wu9121@126.com (Z.-L.W.); chensysau@163.com (S.-Y.C.); jaxb369@sicau.edu.cn (X.J.); fdeng@uark.edu (F.D.); wjie68@163.com (J.W.); 2Liangshan Kehua Dairy Cow Breeding Co., Ltd., Xichang 615000, China

**Keywords:** dairy cow, clinical ketosis, periparturient, transcriptome

## Abstract

Ketosis is one of the most prevalent transition metabolic disorders in dairy cows, and has been intrinsically influenced by both genetic and nutritional factors. However, altered gene expression with respective to dairy cow ketosis has not been addressed yet, especially at the genome-wide level. In this study, we recruited nine Holsteins diagnosed with clinical ketosis and ten healthy controls, for which whole blood samples were collected at both prepartum and postpartum. Four groups of blood samples were defined: from cows with ketosis at prepartum (PCK, N = 9) and postpartum (CK, N = 9), respectively, and controls at prepartum (PHC, N = 10) and postpartum (HC, N = 10). RNA-Seq approach was used for investigating gene expression, by which a total of 27,233 genes were quantified with four billion high-quality reads. Subsequently, we revealed 75 and four differentially expressed genes (DEGs) between sick and control cows at postpartum and prepartum, respectively, which indicated that sick and control cows had similar gene expression patterns at prepartum. Meanwhile, there were 95 DEGs between postpartum and prepartum for sick cows, which showed depressed changes of gene expression during this transition period in comparison with healthy cows (428 DEGs). Functional analyses revealed the associated DEGs with ketosis were mainly involved in biological stress response, ion homeostasis, AA metabolism, energy signaling, and disease related pathways. Finally, we proposed that the expression level of *STX1A* would be potentially used as a new biomarker because it was the only gene that was highly expressed in sick cows at both prepartum and postpartum. These results could significantly help us to understand the underlying molecular mechanisms for incidence and progression of ketosis in dairy cows.

## 1. Introduction

Dairy cows, especially for high-production individuals, are very sensitive to metabolic diseases and environmental stresses during the periparturient period, because they are apt to suffer from negative energy balance (NEB) [1]. NEB in early-lactating cows is characterized by a failure of hepatic gluconeogenesis to supply adequate glucose for maintenance and lactation [2]. If the NEB-induced metabolic disorders could not be properly adapted and addressed, cows would finally develop into subclinical or clinical ketosis [3]. Therefore, ketosis is the most common metabolic disease in dairy cows during the periparturient period, with reported prevalence ranging from 6.9% to 43% [3,4,5]. In practice, ketosis can significantly decrease milk production and reproduction efficiency, and also increase the risk of displaced abomasum, lameness, and metritis [3,6].

Ketosis can be clinically indicated by elevated blood concentrations of ketone bodies, such as β-hydroxybutyrate (BHBA), acetoacetate, and acetone [7,8,9]. Many studies have found that BHBA is a predominant and stable blood ketone body in ruminant ketosis [4,10], which, therefore, has been widely used for clinically diagnosing and classifying ketosis in dairy cows [11]. Furthermore, a large number of molecular biomarkers have been also identified to be associated with ketosis, such as milk fatty acids [12], serum hepatokines [13], inflammatory biomarkers [14], methylglyoxal [2], metabolites [15], mineral elements [16], protein profiling [17], and amino acids [18] in blood.

Studies found significant genetic corrections between ketosis and other health events [19], and genetic selection for improved resistance to ketosis is feasible [20]. Nevertheless, the association between difference in gene expression and incidence and progression of ketosis remains unknown. The ‘Omics’ technologies of transcriptomics, metabolomics, and proteomics have increasingly been used to investigate the underlying molecular mechanism of complex diseases in cattle, such as hyperthermic stress [21], mastitis [22], footrot [23], and lipid metabolome disorder [24]. In these studies, whole blood has been widely used due to convenient and non-invasive sampling. Additionally, whole blood is rich with information regarding health conditions, and can represent an alternative to tissue sampling to find molecular signatures of different physiological conditions [25]. In the present study, we employed a whole blood transcriptome approach to investigate the associated alteration of gene expression with clinical ketosis in dairy cows. This research was part of a prospective study designed to elucidate molecular mechanisms and identify predictive biomarkers of clinical ketosis.

## 2. Materials and Methods 

All experimental protocols involved in the present study were approved by the Institutional Animal Care and Use Committee of Sichuan Agricultural University (DKY-B20171906).

### 2.1. Animals and Ketosis Dagnosis

We initially enrolled a total of 74 pregnant individuals at 21 days before due date in a 1500-cow modern dairy farm in Sichuan province, China (Figure 1). All these cows were at third parity with similar due dates and body condition scores (Appendix A). The animals were kept in freestall barns and had free access to fresh water. All 74 cows received the same diet, and the basal formulation is shown in Appendix A.

For each cow, ketosis was diagnosed at both prepartum (−14 days before due date) and postpartum (+14 days after calving) according to blood concentration of BHBA. The plasma BHBA was measured by a hand-held meter TNN (Yicheng, Beijing, China). We stringently defined the occurrence of clinical ketosis cows with BHBA concentration ≥ 2.60 mM and healthy cows with BHBA concentration < 1.00 mM, respectively. Cows were removed from the herd if diagnosed with ketosis at prepartum. In addition, individuals were also excluded if they had other diseases by veterinary examination during the whole experiment period. Finally, a subset of 19 cows from the herd were collected, nine of them with clinical ketosis and 10 healthy controls at postpartum. For the 19 finally collected cows, a total of 38 blood samples were obtained and then divided into four groups, including sick cows at postpartum (CK, N = 9) and prepartum (PCK, N = 9), and healthy controls at postpartum (HC, N = 10) and prepartum (PHC, N = 10) (Figure 1).

### 2.2. Collection of Blood Samples and RNA Extraction

Before the morning feeding, coccygeal vein blood samples were collected into 10 mL vacutainer tubes containing EDTA K2 and quickly stored in liquid nitrogen. Total RNA was isolated from whole blood using TRIzol Reagent (TaKaRa, Dalian, China) according to the standard protocol. DNA was cleaned out using the RNeasy Midi Kit (Qiagen, Valencia, CA, USA) with DNase digestions. RNA purity and concentration were measured using NanoPhotometer^®^ spectrophotometer (IMPLEN, Westlake Village, CA, USA) and Qubit^®^ RNA Assay Kit in Qubit^®^ 3.0 Flurometer (Life Technologies, Carlsbad, CA, USA), respectively. RNA integrity was assessed using the RNA Nano 6000 Assay Kit of the Agilent Bioanalyzer 2100 system (Agilent Technologies, Santa Clara, CA, USA). RNA quality was verified by ensuring all RNA samples had an absorbance (A260/280) of between 1.80 and 2.06, and RNA integrity number of between 7.7 and 9.7.

### 2.3. Library Preparation and Sequencing

According to manufacturer’s recommendations, RNA-Seq libraries were constructed with approximately 1 μg RNA per sample using NEBNext^®^ UltraTM RNA Library Prep Kit of Illumina^®^ (NEB, Ipswich, MA, USA). In brief, mRNA was purified from total RNA using poly-T oligo-attached magnetic beads, and fragmentation was carried out using divalent cations under elevated temperature. First-strand cDNA was synthesized using random hexamer primer and M-MuLV Reverse Transcriptase (RNaseH-), and second-strand cDNA synthesis was subsequently performed using DNA Polymerase I and RNase H. Remaining overhangs were converted into blunt ends via exonuclease/polymerase activities. After adenylation of 3′ ends of DNA fragments, NEBNext Adaptor with hairpin loop structure was ligated for hybridization. Finally, products were purified (AMPure XP system) and library quality was assessed on the Agilent Bioanalyzer 2100 System. The library preparations were sequenced on an Illumina Hiseq X Ten platform, and 150 bp paired-end reads were generated.

### 2.4. Reads Mapping and Quantification of Gene Expression

All raw reads were first subjected to quality control by removing low-quality reads using fastp software (v0.19.8) [26]. The low-quality reads were defined according to one of the following three criterions, including reads containing adaptor sequences, > 10% of ambiguous ‘N’ bases, or > 50% of bases with Phred value ≤ 20. Subsequently, the high-quality reads were aligned to the bovine reference genome (ARS-UCD1.2.95) using HISAT2 software (v2.1.0) with default parameters [27]. The featureCounts tool (v1.5.0-p3) [28] was employed to calculate the number of mapped reads to each gene. After filtering out genes with less than one raw count in average, principal component analysis (PCA) was further applied using the plotPCA function in DESeq2 R package (v1.22.2) [29]. For exploring the distribution of read counts for each group, the expression density was carried out through a density plot.

### 2.5. Differentially Expressed Genes and Functional Enrichment

The differentially expressed genes (DEGs) among different groups were analyzed using DESeq2 R package (v1.22.2). DESeq2 provides statistical analysis for determining DEGs using the negative binomial distribution model. The *p*-values were adjusted (padj) using the Benjamini-Hochberg approach for controlling the false discovery rate [30]. Finally, both padj < 0.05 and |log2(FoldChange)| > 1 were set as the threshold for defining DEGs [31]. The pheatmap package (v 1.0.12) was further used for visualizing these DEGs. To evaluate the functional implication of DEGs, we subsequently performed Gene Ontology (GO) enrichment and Kyoto Encyclopedia of Genes and Genomes (KEGG) pathway analysis with clusterProfiler R package (v3.10.1) [32] with a Benjamini-Hochberg-adjusted *p*-value (padj) of < 0.05.

### 2.6. Validation of RNA-Seq Data by qPCR

To verify the repeatability and reproducibility of DEGs obtained from RNA-Seq, eight DEGs were chosen for qPCR validation. The qPCR primers (Appendix A) were designed using Primer Premier 5.0 software based on consensus cDNA sequence of each gene. All the 38 RNA samples used in RNA-Seq analyses were used to prepare cDNA. Single stranded cDNA was synthesized from 1.5 μg of RNA using a PrimeScript RT reagent kit (TaKaRa, Dalian, China). qPCR was performed on Bio-Rad CFX96 real-time PCR detection system (Bio-Rad, Inc., Hercules, CA, USA). The expression level of genes was normalized to *GAPDH*. Relative gene expression levels were calculated using the 2^−ΔΔCT^ method [33].

## 3. Results

### 3.1. BHBA Parameters and Ketosis Diagnosis

For the 19 finally collected cows, the blood BHBA concentration at both prepartum and postpartum are presented in Figure 2. At prepartum, all cows had < 1.0 mM BHBA and did not differ statistically significant between PHC and PCK groups. At postpartum, the CK group had a plasma BHBA concentration of 2.79 ± 0.12 mM, and HC group with 0.65 ± 0.22 mM on average.

### 3.2. Gene Expression and Cluster Analysis

A total of 3.88 billion clean reads were successfully obtained with an average of 102.2 million per sample, and 89% of them could be uniquely mapped to reference genome (Appendix A). A total of 27,233 annotated genes were quantified, and we detected 17,543 expressed genes among these samples after removing the genes with less than one raw count in average. According to the gene expression levels, all samples were first clustered using PCA (Figure 3a). The first two principle components accounted for 39% of total variance. Beside HC group, all cows could be relatively separated and clustered together for cows in CK, PCK, and PHC groups. Subsequently, we drew a density plot (Figure 3b) and showed that the patterns of normalized read count distribution for the expressed genes were similar.

### 3.3. Differential Expression of Genes and Validation

Between CK and HC groups, a total of 75 genes were detected to be differentially expressed, 27 genes had higher relative expression level in the CK group than in the HC group, and another 48 genes significantly decreased in the CK group. (Figure 4a and Appendix A). However, only four genes, syntaxin 1A (*STX1A*), striated muscle enriched protein kinase (*SPEG*), *ENSBTAG00000053952*, and *ENSBTAG00000051641*, were differentially expressed when compared to PCK and PHC groups (Figure 4b). Among these four DEGs, *STX1A* was simultaneously significantly more highly expressed in sick cows at both prepartum and postpartum.

To identify the genes responsible for the successful adaption changes during the transition phase, we compared the gene expression between HC and PHC groups. A total of 428 genes were differentially expressed, with 354 genes upregulated and 74 genes downregulated (Appendix A; Figure 4c) postpartum. Furthermore, we found 95 DEGs (Appendix A; Figure 4d) when comparing the CK group to PCK group, which could help to explain that their dysregulation contributes to ketosis from prepartum to postpartum.

To validate RNA-Seq data, a total of eight DEGs were selected for qPCR analysis. Among them, three genes were differentially expressed between PCK and PHC groups and five genes were differentially expressed between CK and HC groups. The results showed that the trends of gene expression were concordant between RNA-Seq and qPCR results (Appendix A).

### 3.4. Functional Enrichment Analysis of DEGs

To explore the biological implication of DEGs associated with ketosis, enrichment analyses of both GO terms and KEGG pathways were performed. For the 75 DEGs between CK and HC groups, multiple GO terms and pathways were significantly enriched (Figure 5 and Appendix A), which were mainly related to biological stress response (such as “response to external stimulus and endogenous stimulus”) and ion homeostasis (such as “iron ion homeostasis” and “cation and inorganic homeostasis”). Moreover, we found GO terms related to myeloid cell and erythrocyte homeostasis, including “myeloid cell development and differentiation”, “hemopoiesis”, “erythrocyte differentiation”, and “erythrocyte and myeloid cell homeostasis”. In addition, KEGG pathway analysis showed that nine pathways were significantly enriched, and several ones were associated with AA metabolism (arginine and proline metabolism), energy signaling (AGE-RAGE signaling pathway in diabetic complications), and disease-related pathways (proteoglycans in cancer, amoebiasis, and focal adhesion).

For the DEGs between HC and PHC groups, we observed the enriched GO terms of “calcium ion binding”, “lipase and phospholipase activity”, “organic acid and lipid acid binding”, and KEGG pathways of “IL-17 signaling pathway” and “NOD-like receptor signaling pathway” (Appendix A). In contrast to healthy cows, the enriched GO terms of DEGs between CK and PCK were showed to be mainly related to ion homeostasis; stress responses and lipid metabolic process; and KEGG pathways of “Transcriptional misregulation in cancer”, “Hematopoietic cell lineage”, “Malaria”, and “Amoebiasis” (Appendix A).

## 4. Discussion

Ketosis is a common metabolic disorder in dairy cows and can lead to enormous economic losses by decreasing milk production, impairing reproductive performance, and increasing other diseases [34,35,36,37]. At early lactation, cows are apt to suffer from NEB due to milk production and limited feed intake [38]. At the same time, massive fat mobilization from different adipose depots leads to fatty acid β-oxidation in hepatic mitochondria to produce energy, but overload fatty acid occurs in consequence of the elevated concentrations of ketone bodies and fatty liver syndrome [2,39,40]. Previous reports have showed that elevation of BHBA in the systemic circulation contribute to metabolic acidosis [41], and metabolic acidosis is associated with minerals elements metabolism [16,42]. The standard diagnostic test for ketosis is the concentration of blood BHBA, for which the varied cut-point values, ranging from 1.0 mM to 1.4 mM [5,43,44], were used. The higher cut-point values of BHBA were used for diagnosing clinical ketosis, such as 1.6 mM and 1.8 mM [45,46]. In the present study, we conservatively employed 2.6 mM BHBA as cut-point value for defining the cows with clinical ketosis because of two considerations. First, we tried to avoid the false-positive of clinical ketosis by increasing the cut-point value. Second, it had been observed that cows with blood BHBA ≥ 2.6 mM have an obvious reduction of dietary intake and milk production in our farm according to former field experiences.

In this study, GO and KEGG analyses identified that multiple pathways in relation to ion homeostasis were dysregulated when ketosis occurred. This finding is consistent with the previous report that irreversible loss of minerals may result in hypocalcemia in cattle [47,48]. We hypothesized that mineral elements alterations may be one of the complications to inducing ketosis. In addition, enrichment of GO terms such as myeloid cell and erythrocyte homeostasis was detected, which would indicate that ketosis caused the systemic disease in dairy cow [49]. Furthermore, some of the DEGs could be linked to disease-related pathways, such as proteoglycans in cancer, amoebiasis, and focal adhesion. It is possible that these candidate genes play important roles in ketosis occurrence. Future studies will better elucidate relationships between the altered expression level and ketosis.

It has previously been reported that periparturient is characterized by dramatic and sudden physiological changes [50]. However, animals differ tremendously in their adaptive success [51]. Results from HC group versus PHC group showed that the DEGs were primarily involved in ion homeostasis, fat digestion, and lipid metabolism; this is a finding that is similar to what has been reported using metabolomic analysis in transition cows [52]. Otherwise, the inflammation-related pathways of “IL-17 signaling pathway” and “NOD-like receptor signaling pathway” were obtained. Some degree of inflammation might even be required for successful adaptation [51]. It should be noted that there were more GO terms and KEGG pathways obtained from CK group versus PCK group; the dysregulation of such pathways may induce poor adaptive response during transition period and promote ketosis. Future work is planned to determine if the same is observed in larger populations.

In order to explore the global gene expression profile and candidate genetic biomarkers at prepartum, a comparison between PCK and PHC groups was carried out. A total of four DEGs was screened out, and all of them were expressed more in animals with ketosis, which showed all animals had a similar gene expression pattern at prepartum. Notably, *STX1A* was screened out at both prepartum and postpartum. STX1A as a presynaptic protein is widely expressed in brain, endocrine system, heart, and other organs [53]. *STX1A* has been associated with myocardial ischemia-reperfusion by regulating KATP and calcium channels signaling pathways [54]. Moreover, *STX1A* mediates isoflurane-induced alleviation of hypoxia-reoxygenation injury in rat cardiomyocytes [55]. The expression level of *STX1A* is associated with stimulation such as stress, ischemia, and hypoxia/reoxygenation in cardio disease [56]. These findings suggest a potential role of *STX1A* in environmental stress and ion homeostasis, which plays important role in the ketosis process. It may be considered as useful as screening, diagnostic, and predictive biomarkers of ketotic cows. However, the potential biological effect of the *STX1A* on cow ketosis requires further investigation.

## 5. Conclusions

In summary, we systematically investigated the differentially expressed genes and pathways associated with transition dairy cows, and revealed that multiple pathways in relation to biological stress response, ion homeostasis, AA metabolism, energy signaling, and disease-related were altered when ketosis occurred. Additionally, the expression level of *STX1A* was suggested to be one potential molecular marker for predicting the incidence of ketosis at prepartum.

## Figures and Tables

**Figure 1 genes-11-00219-f001:**
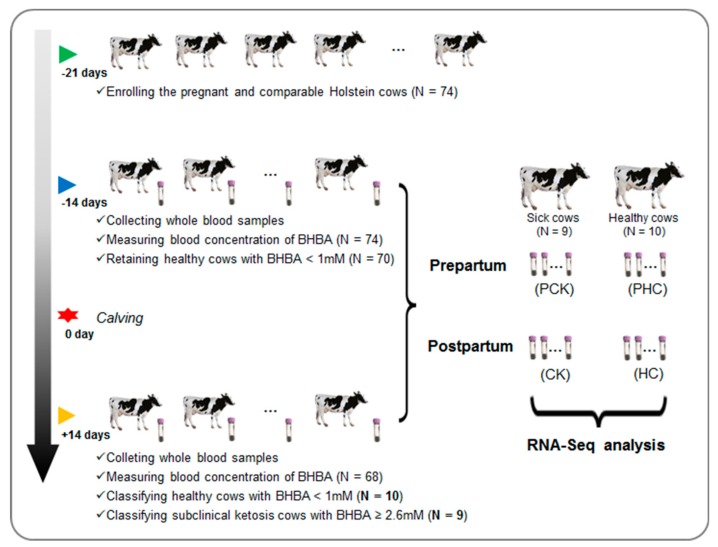
An overview of the experiment design and sample collection.

**Figure 2 genes-11-00219-f002:**
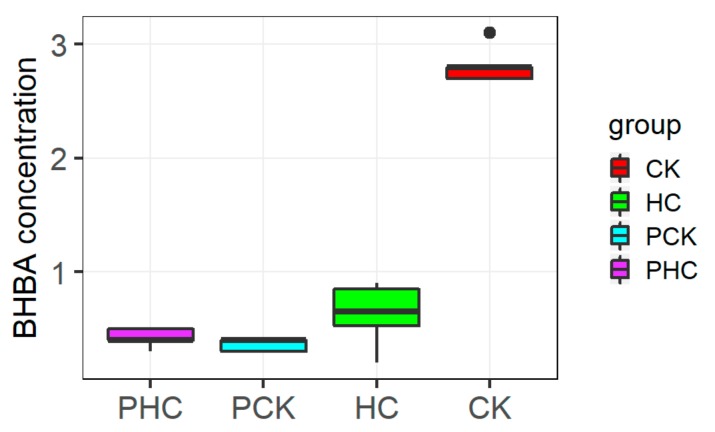
The measured β-hydroxybutyrate (BHBA) values among different groups.

**Figure 3 genes-11-00219-f003:**
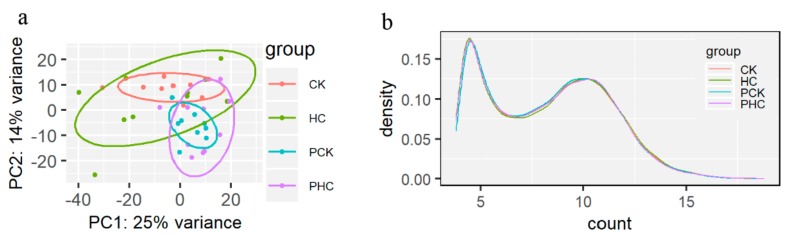
Gene expression patterns and gene level expression abundance among the four groups. (**a**) The principal component analysis (PCA) plot of transformed read counts for each group and (**b**) the density plot of transformed read counts for each group.

**Figure 4 genes-11-00219-f004:**
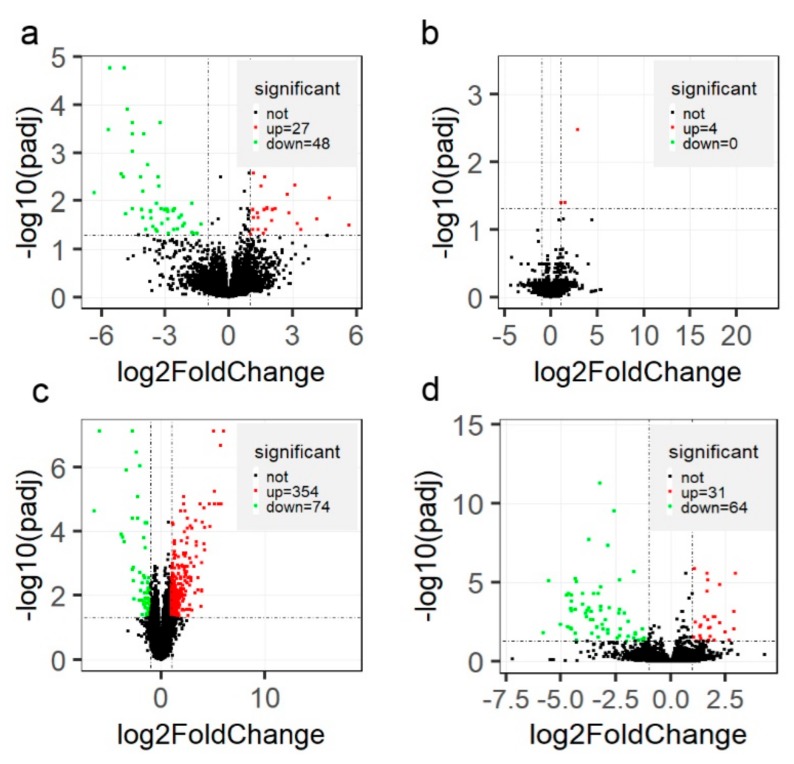
Volcano plot of differentially expressed genes (DEGs) (padj < 0.05 and |log_2_(FoldChange)| > 1) among different groups. The x-axis represents the log_2_(FoldChange), while y-axis represents statistical significance for each gene. The pairwise comparisons are ketosis at postpartum (CK) versus healthy controls at postpartum (HC) (**a**), ketosis at prepartum (PCK) versus healthy controls at prepartum (PHC) (**b**), HC versus PHC (**c**), and CK versus PCK groups (**d**), respectively.

**Figure 5 genes-11-00219-f005:**
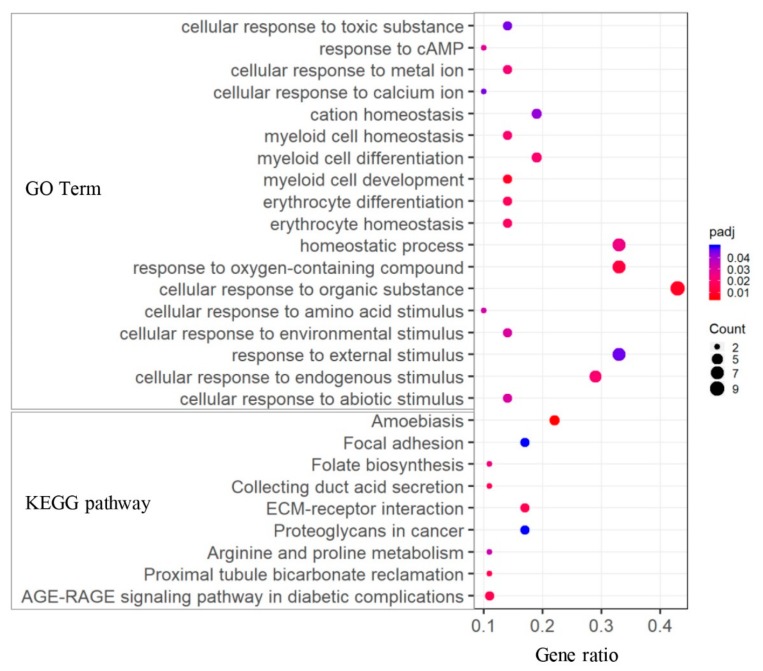
GO and KEGG analyses of DEGs between CK and HC groups.

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
