# Peer review of "Clinical Ketosis-Associated Alteration of Gene Expression in Holstein Cows"

_genes, 2020, doi:10.3390/genes11020219_

Round 1

Reviewer 1 Report

Brief summary:

This paper presents a study that investigated association between difference in gene expression and clinical ketosis in dairy cows from a systematic genome-wide assessment of whole blood transcriptomes. Beta-hydroxybutyrate (BHBA) was examined in 74 Holstein dairy cows of 3rd parity prepartum (-14 days before due date) and postpartum (14 days after calving). None of these had ketosis (defined by BHBA concentration >= 2.60mM) prepartum. A group 9 cows had ketosis postpartum (without other clinical disease). Another group of 10 cows had BHBA<1.0 both pre- and postpartum and no others disease. Gene expressions were compared within group pre-/postpartum and between groups prepartum and postpartum. A number of differentially expressed genes were found to be associated with ketosis and related to multiple pathways.

Broad comments:

Every time I read “CK” my first thought was “controls”. Why not more simply write “K” (and “PK”) for ketosis group (and prepartum ketosis group) and “C” for (healthy) control group (and PC for prepartum control group). Note also that “Abbreviations should be defined in parentheses the first time they appear in the abstract, main text, and in figure or table captions and used consistently thereafter.” (c.f. Instructions for Authors, https://www.mdpi.com/journal/genes/instructions). At least DEG should be defined in the abstract. I am wondering if 9 and 10 cows is a large or small sample? Is the sample size a strength or a weakness of your study? The manuscript was generally easy to read – though I have noted a few places that was difficult to read in the specific comments.

Specific comments:

L17: I suggest deleting “the”.

L19-20: It seems a bit strange to write postpartum before prepartum.

L18-20: I think it is a bit clearer to write something like: “Four groups of blood samples were define: from cows with ketosis prepartum (PCK, N=9) and postpartum (CK, N=9) respectively from controls prepartum (PHC, N=10) and postpartum (HC, N=10).”

L25: “the depressed” I suggest deleting ‘the’.

L37: I suggest deleting last ‘the’.

L38-39: Mismatch between past and present tense.

L38: “the NEB-induced” ... are you certain about this causality?

L38-39: If ketosis is itself a metabolic disease then what is the difference between this and the ‘metabolic disorders’ leading to ketosis?

L40: I suggest deleting the last ‘the’

L41: A prevalence is normally a proportion (though often wrongly referred to as a rate). To avoid this problem you could simply write ‘prevalence’ instead of ‘prevalence rates’.

L44: I suggest deleting ‘the’.

Ref. 7: Please find an updated reference that is no “just” in a conference proceedings. If it is such a high prevalent disease, I am sure someone has written a paper over the last 12-13 years.

L49: “metabolites”  ... in milk? in blood? both???

L51-52: I do not understand this sentence – either elaborate or rewrite. More precisely, what do you mean by “variability has been found in resistance”? Moreover, I suggest deleting “the” before “genetic selection”.

L52-53: Rewrite – it is almost unreadable.

Section 2.1: I think you can make this a bit more precise. It appears that the BHBA threshold to define ketosis cases is applied postpartum. Would you also include cows with BHBA above the threshold prepartum in this group, had there been some? It also appears that you have had some exclusion criteria as the cow with animal ID 8566 also had BHBA above the threshold but is noted to have mastitis. For controls, it appears the threshold criteria had to be fulfilled both pre- and postpartum ... and in addition an exclusion criteria of any disease. I first wondering why you chose 10 controls (why not 9 to make the design balanced?) – but after looking at supplementary table S1 it becomes clear the these 10+9 animals were ALL animals fulfilling thresholds and not having other diseases noted. I think you should make this clear.  

Table S2: The percentages under the “Nutrient component” header adds to much more than 100% (i.e. this is not just due to rounding). Something must be wrong ...

L108: “fastp” should this be ‘FASTQ’ (q instead of p and maybe capitalised)?

L109: “one of the three criterions,” maybe ‘one of following three criterions:’

Section 2.4: Why are some software/packages mentioned with a reference and some not? Are there no references for DESeq2 and ggplot2? Ah, DESeq2 is referenced in section 2.5 – I suggest moving this reference to section 2.4.

L115: I suggest changing “analyzed” to ‘applied’.

L122: Ref. 28 ... why not use a more direct reference to Benjamini & Hochberg? Moreover, “Bonferroni-corrected”??? you just said that you used Benjamini and Hochberg’s approach for adjustment of p-values.

L138: “differ statistically” ... I would write ‘differ statistically significant’.

L139-142: The postpartum testing of BHBA between CK and HC is more or less stupid ... per construction CK cows are above 2.60 and HC cows are below 1.0 – no need to test that they differ! Moreover, since all cows were <1.0 prepartum this construction also ensures that ALL CK cows will increase more than EVERY single one of the HC cows: HC cows will have an increase <1.0; CK cows will have an increase >1.6.

L147: “89.38%” is it not overkill to show two decimals?

L147-148: Either change the comma to ‘and’ or write something like ‘From a total of ..., we detected ...’

L169: Maybe to make it crystal add ‘postpartum’ after “downregulated”.

Figure 4: I can figure that the vertical and horizontal lines indicated statistical significantly up-/downregulated groups but how are these defined? Ah, you actually mention this in methods ... I think you should consider restating that DEGs are those with corrected p<0.05 (i.e. –log10(adj. p)>1.3) and |log2(FoldChange)|>1.

Figure S1: Would not harm to make this bigger.

L182: “the enrichment analyses” ... delete “the”.

L184: “Multiple” does this have to be capitalised?

Figure 5: I think you need to explain the measures; “GeneRatio” (why not ‘Gene ratio’?), “Count” and “padj”.

Figure 6: I actually like the plain text description but everywhere else in the paper, you refer to PHC and HC (fig. 6a) and PCK and CK (fig. 6b). Why is the axis of fig 6b extended far beyond the largest gene ratio? Moreover, you might want to make readers aware that all three scales are different between plots, i.e. x-axes, count bobbles and padj colour-bars are NOT comparable between the plots in fig 5, 6a and 6b.

L221: I suggest deleting “the”.

L224: I suggest changing “the ketosis occurrence” to ‘ketosis occurred’ or changing “when the ketosis occurrence” to ‘with ketosis occurrence’.

L226: I suggest changing “to induce” to ‘inducing’.

L227: I suggest deleting “the” and probably also the comma after “terms”.

L231-232: How? How do you know?

L233: I suggest inserting ‘that’ after “reported”.

L244: I suggest deleting “the” or change it to ‘a’.

L245: “sick animals” maybe more precise: ‘animals with ketosis’.

L244-246: I do not understand your conclusion (or how you get to this conclusion from the first part of the sentence).

L247-250: Quite long sentence and a bit difficult to follow. Could it be divided into two?

L250: Consider changing “was” to ‘is’ as this is not a result from your study.

L252-253: Consider changing “suggested” to ‘suggests’, delete “and” before “which” and change “played” to ‘plays’.

L274: Delete hyphen between “writing” and “original”.

Author Response

Dear Reviewer,

Special thanks to you for your good comments!

Those comments are all valuable and very helpful for revising and improving our paper, as well as the important guiding significance to our researches. We have checked the manuscript and revised it according to the comments. We submit here the revised manuscript as well as a list of changes. We appreciate for the your warm work earnestly, and hope that the correction will meet with approval.

Once again, thank you very much for your comments and suggestions!

Sincerely,

Songjia Lai

Reviewer 2 Report

The research was properly designed and reported. 

Line 239 'note' should be changed to 'noted'

Author Response

Dear Reviewer,

Thank you for your comments!

Line 239 'note' should be changed to 'noted'

Response: L239, “note” was corrected as “noted”.

We tried our best to improve the manuscript and made minor changes in the revised version. And here we did not list the all the changes but clearly highlighted by using the "Track Changes" function in Microsoft Word. We have corrected some expression errors, these changes will not influence the content and framework of the paper. We appreciate for reviewers’ warm work earnestly, and hope that the correction will meet with approval.

Once again, thank you very much for your comments.